# Prognosis of older patients with newly diagnosed AML undergoing antileukemic therapy: A systematic review

Qiukui Hao [1,2,3]*, Farid Foroutan[3], Mi Ah Han[4], Tahira Devji[3], Fernando Kenji Nampo[5], Sudipto Mukherjee[6], Shabbir M. H. Alibhai[7], Ashley Rosko[8], Mikkael A. Sekeres[6], Gordon H. Guyatt[3], Romina Brignardello-Petersen[3]*

1 The Center of Gerontology and Geriatrics (National Clinical Research Center for Geriatrics), West China Hospital, Sichuan University, Chengdu, China, 2 School of Rehabilitation Science, McMaster University, Hamilton, Ontario, Canada, 3 Department of Health Research Methods, Evidence, and Impact, McMaster University, Hamilton, Ontario, Canada, 4 Department of Preventive Medicine, College of Medicine, Chosun University, Gwangju, Republic of Korea, 5 Latin-American Institute of Life and Nature Sciences/Evidence-Based Public Health Research Group, Federal University of Latin-American Integration, Foz do Iguassu, Brazil, 6 Leukemia Program, Cleveland Clinic, Cleveland, Ohio, United States of America, 7 Department of Medicine, Princess Margaret Cancer Centre, University Health Network, Toronto, Canada, 8 Division of Hematology, The Ohio State University, Columbus, Ohio, United States of America

* haoqiukui@gmail.com (QH); brignarr@mcmaster.ca (RBP)

**Data Availability Statement:** All relevant data are within the paper and its Supporting information files.

## Abstract

### Background and objective

The prognostic value of age and other non-hematological factors in predicting outcomes in older patients with newly diagnosed acute myeloid leukemia (AML) undergoing antileukemic therapy is not well understood. We performed a systematic review to determine the association between these factors and mortality and health-related quality of life or fatigue among these patients.

### Methods

We searched Medline and Embase through October 2021 for studies in which researchers quantified the relationship between age, comorbidities, frailty, performance status, or functional status; and mortality and health-related quality of life or fatigue in older patients with AML receiving antileukemic therapy. We assessed the risk of bias of the included studies using the Quality in Prognostic Studies tool, conducted random-effects meta-analyses, and assessed the quality of the evidence using the Grading of Recommendations, Assessment, Development and Evaluation approach.

### Results

We included 90 studies. Meta-analysis showed that age (per 5-year increase, HR 1.16 95% CI 1.11–1.21, high-quality evidence), comorbidities (Hematopoietic Cell Transplantation-specific Comorbidity Index: 3+ VS less than 3, HR 1.60 95% CI 1.31–1.95, high-quality evidence), and performance status (Eastern Cooperative Oncology Group/ World Health Organization (ECOG/WHO): 2+ VS less than 2, HR 1.63 95% CI 1.43–1.86, high-

**Funding:** The authors received no specific funding for this work.

**Competing interests:** The authors declare no competing financial interests.

quality evidence; ECOG/WHO: 3+ VS less than 3, HR 2.00 95% CI 1.52–2.63, moderate-quality evidence) were associated with long-term mortality. These studies provided inconsistent and non-informative results on short-term mortality (within 90 days) and quality of life.

### Conclusion

High-quality or moderate-quality evidence support that age, comorbidities, performance status predicts the long-term prognosis of older patients with AML undergoing antileukemic treatment.

### Introduction

Acute myeloid leukemia (AML) commonly occurs in older adults. The median age at diagnosis is 68 years or 72 years in the United States or Sweden, respectively [1, 2]. Even though both supportive care and antileukemic therapies have improved over the past several decades, the overall survival remains dismal, particularly in certain subgroups. Universally, more than half of patients with AML die from their disease [3]. Among patients aged 65–75, less than 30% survive more than 2 years, and among patients aged 75 years or older, less than 20% survive more than 2 years, with patients 85 years and older having the lowest survival rates [2, 4].

The best management strategy for older patients with AML is still unclear. Formal geriatric or frailty assessments are recommended before deciding between intensive or non-intensive therapy or opting for best supportive care alone [5, 6]. Increasing age is strongly associated with the presence of comorbidities and functional impairment in older cancer patients [7]. In patients with AML, fatigue is one of the common symptoms and is related to quality of life [8]. Even though older patients with AML may have similar quality of life (QOL) between intensive and non-intensive chemotherapy [9], QOL is an important outcome to consider for treatment decisions [10] and clinical trials [11]. Numerous non-hematological factors are thought to influence the treatment choice for older adult with AML, including performance status or functional level, comorbidities, and age [6].

Even the influence of advanced age on outcomes within older AML patients is controversial: one retrospective study evaluated consecutive older adults (60 years or older) with AML and myelodysplastic syndromes who underwent allogeneic hematopoietic stem cell transplant and found that age was not associated with outcomes such as non-relapse mortality, relapse, relapse-free survival and overall survival [12]. In contrast, researchers from Germany conducted a post-hoc analysis of a randomized controlled trial including subjects aged 57 to 63 years and concluded that age, not interventions, determines outcomes of adults with AML [13]. Comorbidity and performance/functional status in older patients, which are remarkably heterogeneous, have also been suggested as potentially helpful variables to identify older patients at risk for particularly poor outcomes [14–16].

The prognostic value of age, comorbidities, performance status (PS) and functional status, in predicting outcomes remains unclear among older patients with AML receiving antileukemic therapy [12, 13, 17]. Currently, there is no systematic review addressing this issue. Therefore, we conducted a systematic review to determine the association between these prognostic factors and mortality and health-related quality of life or fatigue of older patients with newly diagnosed acute myeloid leukemia (AML) undergoing antileukemic therapy.

## Materials and methods

We conducted a systematic review of the literature to provide evidence to support the 2020 American Society of Hematology (ASH) guidelines for the management of older patients with AML [18]. We did not register a protocol, but followed methods agreed by the expert panel a priori.

### Eligibility criteria

We included studies meeting all the following criteria: (1) patients with newly diagnosed AML, including de novo AML, treatment-related AML, and secondary AML; (2) studies that enrolled participants 55 years or older; studies that enrolled participants of any age, but reported results for those 55 years or older or other higher age cut-off points (e.g. 60, 65, 70 years) separately; or studies that enrolled participants of any age that reported results combined for all, but in which 75% or more of the participants were 55 years or older; (3) patients who received any type of antileukemic therapy, including intensive or low-intensity induction antileukemic therapy with any agent, followed by any type of management option for post remission/ consolidation/ maintenance therapy; (4) researchers assessed the prognostic value for any of the following factors: age, comorbidities, frailty (assessed using any method), performance status (assessed using the Eastern Cooperative Oncology Group (ECOG) Performance Status, Zubrod, WHO Performance Status, Karnofsky Score, or any other method, as long as the researchers called it performance status), functional status (assessed by the patient using any instrument, or with any objective measure, as long as the researchers called it functional status) in the outcomes mortality, quality of life (QoL) or fatigue (QoL and fatigue must have been reported by the patient, using any scale or instrument for measurement); (5) studies in which researchers included all consecutive patients who met the eligibility criteria and the researchers performed multivariable or univariate regression models to quantify the influence of any of the factors mentioned above in mortality, quality of life or fatigue. We group all eligible studies for syntheses according to the reported outcomes and the cut-off values of the above prognostic factors.

We excluded studies in which patients had acute promyelocytic leukemia or myeloid proliferations related to Down syndrome, patients received only palliative/ supportive care, or treatment with hydroxyurea alone; studies in which the loss to follow up was over 40%; studies in which researchers included less than 50 patients in the analysis; and studies published only in abstract format.

### Data sources and searches

We searched MEDLINE and EMBASE from inception through October 2021. The ASH AML guidelines considered evidence published through January, 2020. The search strategies are presented in S1 Appendix. We did not limit by language of publication.

### Study identification, data abstraction, and risk of bias assessments

Pairs of reviewers (QH, MAH, TD, FKN) independently screened titles and abstracts of identified references and assessed the full text of all potentially eligible studies, and conducted data abstraction and risk of bias assessments.

Extracted data included study design, the time frame of recruitment, population characteristics, interventions, interested factors and outcomes, statistical methods, estimates of effect and confidence interval. We contacted authors for missing information needed for pooled analyses, which include point-estimates, 95% confidence interval, units, and category of

predictors. If the same study was reported in more than one publication, we used the most comprehensive report. If the participants were the same, we used the longest follow-up period.

We assessed the risk of bias of individual studies using the Quality in Prognostic Studies tool (QUIPS), which includes the assessment of the following domains: patient selection process, study attrition, prognostic factors measurement, outcome measurement, study confounding, and statistical analysis and reporting [19]. Studies with five or more domains judged at low-risk of bias were considered at low-risk of bias. Studies with two or more domains at high-risk risk of bias were considered at high-risk of bias. Otherwise, we judged the studies at an moderate risk of bias.

We trained and calibrated reviewers at all stages. Reviewers solved disagreements at all stages by discussion or consultation with a methodologist or hematologist in our review team.

## Data synthesis and statistical analysis

We defined the short-term mortality as treatment-related mortality (original study defined) or death within 3 months (90 days). Long-term mortality was defined as mortality after 3 months.

Eligible studies reported the prognostic value of assessed factors in predicting outcomes of interest using Hazard ratio (HR), Odds ratio (OR) and risk ratio (RR), and their respective 95% CIs as the measure of effect. We pooled studies separately according to this measure effect (HR, OR and RR), and pooled HR estimates for each predictor as the primary outcome using DerSimonian and Laird random-effects models meta-analyses. All statistical analyses were conducted using the *Meta* package [20] in software R 3.6.0 (R Core Team, 2019).

We assessed heterogeneity through observation of point estimates consistency and overlap of 95% CIs, but not by $I^2$ statistics because of performance issues in prognostic studies with precise estimates and large sample sizes [21].

## Assessment of quality of the evidence

We used the Grading of Recommendations, Assessment, Development and Evaluation (GRADE) approach to assess the overall quality of evidence, which could be rated as high, moderate, low, or very low [21]. We considered concerns related to risk of bias, imprecision, inconsistency, indirectness, and publication bias. We used funnel plots and Egger's test to examine the possibility of publication bias when the number of studies in a meta-analysis was 10 or over.

## Subgroup and sensitivity analyses

We performed subgroup analyses to explore the influence of risk of bias, intervention type and recruitment time (year) of the first patient into the studies, when the number of studies in a meta-analysis was 10 or over. We stratified the intervention type into intensive and less intensive antileukemic therapy, consistently with the related guideline [18]. We hypothesized that there would the magnitude of prognostic value of the predictors would be larger in studies with high risk of bias, in which patients received intensive antileukemic therapy, and patient recruitment before the year 2000.

## Results

### Study selection and characteristics

The literature search yielded 6,825 references after the removal of duplicate citations. Fig 1 summarizes our search and screening process for eligible studies. We included 90 studies [15,

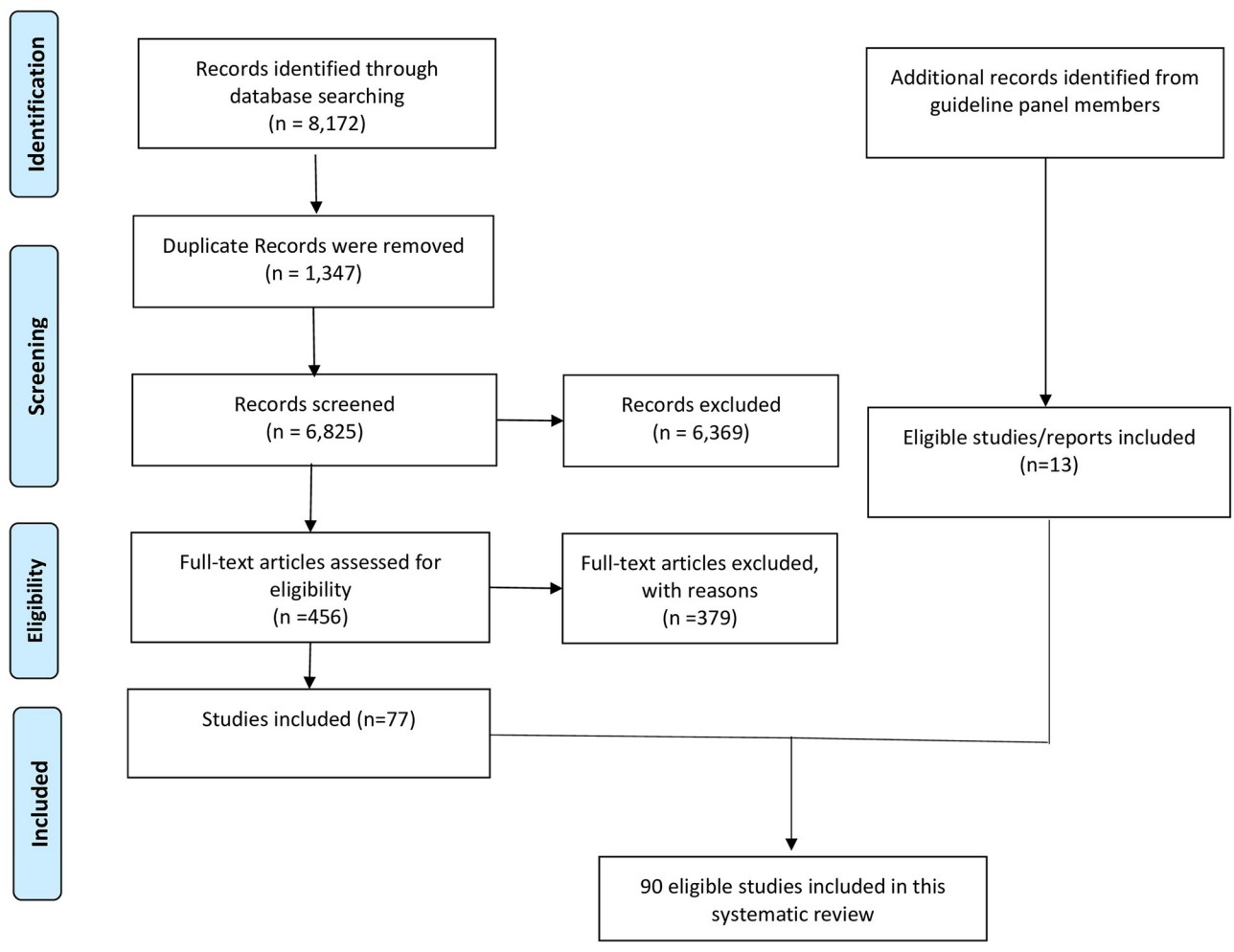

**Fig 1. Flow chart for eligibility assessment according to PRISMA guidelines.**

16, 22–109]. 79 of these studies [15, 16, 22–98] were considered when developing the ASH guidelines. From 24 studies [15, 23, 24, 32, 36, 37, 39, 45, 47, 51, 57, 60, 61, 64, 73, 76, 80, 93–95, 97, 98, 102] that did not report sufficient information to be included in meta- analyses, the authors of three studies responded to queries with additional data [4, 15, 32] In addition, authors from one study provided data about QoL [98].

Overall, 71 studies provided information to be included in meta-analyses. These studies reported various cut-offs for age, used Charlson Comorbidity Index, Hematopoietic cell transplantation-specific comorbidity index, Elixhauser Index or National Cancer Institute comorbidity index to measure comorbidities, and used Eastern Cooperative Oncology Group/WHO performance status (ECOG/WHO), Karnofsky index or Poor Performance Indicators (PPI) to measure performance status. Sixty-nine studies assessed the prognostic value of the factors of interest and long-term mortality: 52 reported on age, 12 on comorbidities, and 37 on performance status. Seven studies assessed the prognostic value of the factors of interest and short-term mortality: four focused on age; four on comorbidities and three on performance status. The characteristics of the 90 eligible studies are summarized in S2 Appendix.

## Risk of bias of individual studies

The risk of bias assessments of individual studies are summarized in S3 Appendix. Of the 90 eligible studies, 17 (18.9%) had low risk of bias [16, 22, 23, 25, 35, 43, 49, 53, 62, 69, 70, 75, 79, 81, 83, 87, 103], 55 (61.1%) had moderate risk of bias [15, 26–28, 30–33, 36–42, 45, 46, 48, 50–52, 54, 56, 58–61, 63, 65–68, 71, 72, 76, 77, 80, 82, 85, 86, 88–90, 92, 94, 96, 98–101, 104, 106–109], 8 (8.9%) had high risk of bias [24, 34, 47, 78, 91, 95, 97, 102] for all predictors and outcomes. The other ten studies had different risk of bias among reported predictors and outcomes: four studies had high risk of bias [55, 57, 73, 84, 93] and one study had moderate risk of bias for age as a predictor of outcomes [44], five studies had high risk of bias for performance status as a predictor of outcomes [29, 73, 74, 93, 105], and one study had high risk of bias for comorbidities as a predictor of outcomes [64]. Most bias was due to study confounding, statistical analysis, and/or reporting issues.

## Effect of prognostic factors on outcomes

Table 1 summarizes the association between prognostic factors and outcomes of interest, and the quality of the evidence.

**Table 1. Grading of recommendations, assessment, development, and evaluation—confidence in estimates of effect for long term mortality.**

| Number of studies (assess with HR) | Quality assessment | | | | | | Effect | Quality |
|---|---|---|---|---|---|---|---|---|
| | Study design | Risk of bias | Inconsistency | indirectness | Imprecision | Other considerations | HR (95% CI) | |
| Age Per 5 years increase | | | | | | | | |
| 22 | Observational studies | Not serious | Not serious | Not serious | Not serious | None | 1.16 (1.11 to 1.21) | ⊕⊕⊕⊕ High |
| Age 60 years old or more VS less than 60 years old | | | | | | | | |
| 2 | Observational studies | Serious | Serious | Not serious | Serious | None | 1.71 (0.97 to 3.03) | ⊕◯◯◯ Very low |
| Age 70 years old or more VS less than 70 years old | | | | | | | | |
| 11 | Observational studies | Not serious | Serious | Not serious | Not serious | None | 1.45 (1.22 to 1.73) | ⊕⊕⊕◯ Moderate |
| Age 75 years old or more VS less than 75 years old | | | | | | | | |
| 6 | Observational studies | Not serious | Not serious | Not serious | Not serious | None | 1.44 (1.22 to 1.68) | ⊕⊕⊕⊕ High |
| CCI: score 1 or over VS less than 1 | | | | | | | | |
| 2 | Observational studies | Not serious | Not serious | Not serious | Serious | None | 0.92 (0.65 to 1.28) | ⊕⊕⊕◯ Moderate |
| HCT-CI: score 2 or over VS less than 2 | | | | | | | | |
| 2 | Observational studies | Not serious | Not serious | Not serious | Serious | None | 1.35 (0.92 to 1.99) | ⊕⊕⊕◯ Moderate |
| HCT-CI: score 3 or over VS less than 3 | | | | | | | | |
| 3 | Observational studies | Not serious | Not serious | Not serious | Not serious | None | 1.60 (1.31 to 1.95) | ⊕⊕⊕⊕ High |
| ECOG/WHO: score 2 or over VS less than 2 | | | | | | | | |
| 21 | Observational studies | Not serious | Not serious | Not serious | Not serious | None | 1.63 (1.43 to 1.86) | ⊕⊕⊕⊕ High |
| ECOG/WHO: score 3 or over VS less than 3 | | | | | | | | |
| 7 | Observational studies | Not serious | Serious | Not serious | Serious | None | 2.00 (1.52 to 2.63) | ⊕⊕⊕◯ Moderate |

## Long-term mortality

**Age.** Twenty-seven studies, reporting on 18,309 patients, provided a HR to quantify the prognostic value of age [15, 16, 25–27, 29, 32, 33, 42, 43, 49, 50, 56, 59, 75, 82, 87, 90, 91, 99–101, 103, 105–107, 109]. Meta-analyses showed that an age increase of 5 year-increments was associated with increased long-term mortality (HR 1.16; 95% CI 1.11 to 1.21, high-quality evidence, Table 1, S4 Appendix). We did not find evidence of subgroup effects for risk of bias. We found that the impact of age was greater in those receiving more intensive antileukemic therapy (HR 1.21; 95% CI 1.14 to 1.28) than in those receiving less intensive therapy (HR 1.08; 95% CI 1.04 to 1.12) (interaction p-value < 0.01). We also found that the impact of age was greater when the first patient was recruited before 2000 (HR 1.28; 95% CI 1.15 to 1.42) than after 2000 (HR 1.13; 95% CI 1.08 to 1.18) (interaction p-value 0.03) (S7 Appendix).

Similar results were found when comparing long-term mortality in patients 60 years or older to those younger than 60 [47, 68]; patients 70 years or older to those younger than 70 [22, 28, 34, 35, 46, 54, 78, 81, 92]; and patients 75 years or older to those younger than 75 [58, 62, 63], and when a cut-off of 65, 68, or 80 years was used to categorize participants [29, 30, 74]. Studies that reported RR or OR as relative effect also showed the similar results [69, 72, 77, 85, 86, 89]. There was no statistical association between age and long-term mortality when researchers used age cut-offs of 66 and 67 years [52, 53] (S4 Appendix).

There were 16 studies that could not be included in the meta-analyses [23, 24, 36, 39, 45, 51, 57, 61, 73, 76, 80, 93–95, 97, 102]. Among these studies, eight reported that age was not statistically significantly associated with long-term mortality (seven of these did not provide a measure of effect) [36, 45, 57, 61, 73, 80, 102] and one study did not report information on category of age [94].

**Comorbidities.** Three studies, including 614 patients, reported a HR for a Hematopoietic Cell Transplantation-specific Comorbidity Index (HCT-CI) score of 3 or more compared to less than 3 in predicting long-term mortality [41, 62, 105]. Meta-analyses showed that a higher HCT-CI score increased the risk for long-term mortality (HR 1.60; 95% CI 1.31 to 1.95, high-quality evidence, Table 1, S5 Appendix).

Similar results were found when assessing the association of long-term mortality and comorbidities when using different tools or cut-off points to measure this risk factor. Only single studies reported results using different tools or cut-off points, including a Charlson Comorbidity Index (CCI) score 2 or more compared to less than 2 [87], a HCT-CI score 4 or more compared to less than 4 [96], or an National Cancer Institute (NCI) comorbidity index score 3 or more compared less than 3 [15] (S5 Appendix). There was no statistical association between comorbidities and long-term mortality when researchers used a CCI cut-off of 1 [63, 83] or an HCT-CI cut-off of 2 [16, 79], or when using the OR as the measure of association [66, 85] (Table 1, S5 Appendix).

There were 3 studies that could not be included in the meta-analysis [64, 80, 109] One study reported that HCT-CI score 3 or more increased the risk for long-term mortality [80]. The second study reported that CCI was not significantly associated with long-term mortality, whereas Myelodysplastic syndromes (MDS)-specific comorbidity index was associated with long-term mortality [64]. The third study reported that Elixhauser comorbidity Index positively associated with the risk for long-term mortality [109].

**Performance status.** Twenty-one studies, including 5,349 patients, reported the HR for performance status measured using ECOG/WHO (score 2 or more compared with less than 2) in predicting long-term mortality [25, 34, 35, 40, 42, 48, 53, 63, 67, 78, 79, 81, 84, 87, 88, 96, 101, 103, 105, 108]. One study separately reported the results of two cohort studies (defined by patients 55 to 65 years and over 65 years) [67]. Meta-analysis showed that ECOG/WHO score

2 or more increased the risk of long-term mortality (HR 1.63; 95% CI 1.43 to 1.86, high-quality evidence, Table 1, S6 Appendix). We did not find evidence of subgroup effects of risk of bias and types of antileukemic therapy and the first patient recruitment time.

Similar results were found when researchers used different tools or cut-off points to measure performance status, including ECOG/WHO score 3 or more compared to less than 3 (HR 2.00; 95% CI 1.52 to 2.63, moderate-quality evidence, Table 1, S6 Appendix) [43, 54, 55, 58, 70, 71, 104], ECOG/WHO score 2 or 3 compared to 0 [38], and the presence of poor performance indicators [15]. There was no statistical association when researchers compared ECOG/WHO score 1 compared to 0 [38] or ECOG/WHO per one-point increase [16] in predicting long-term mortality. Six studies used RR [44, 69, 72, 86] or OR [66, 85] to present the prognostic value of performance status measured by ECOG/WHO with different cut-off points on long-term mortality, which showed the similar results when we included these in the pooled meta-analysis (S6 Appendix).

There were 11 studies that could not be included in the meta-analysis [23, 37, 51, 57, 61, 64, 73, 80, 93, 95, 97]. Among these studies, two studies reported performance status was not statistically significantly associated with long-term mortality [80, 97].

### Short-term mortality

**Age.** Two studies reported a HR for the prognostic value of age with different cut-off points in predicting short term mortality [49, 58]. As with long-term mortality, an increase in age in 5-year increments, or those 75 years or older increased the risk for short-term mortality (HR 1.65; 95% CI 1.08 to 2.53, HR 1.30; 95% CI 1.10 to 1.54, respectively, S1 Table and S4 Appendix). There was no statistical association when researchers used RR for age increase per-5 years on short term mortality [65]. There were 2 studies that could not be included in the meta-analysis and reported that age was significantly associated with short-term mortality [51, 60].

**Comorbidities.** Four studies reported HR [41, 83], RR [65] or OR [66] for the prognostic value of comorbidities measured by CCI or HCT-CI with different cut-off points in predicting short term mortality. These studies provided inconsistent, non-informative results (S1 Table and S5 Appendix).

### Performance status

Two studies reported that an ECOG/WHO score of 2 or more increased the risk for short-term mortality [49, 58]. Similar results were found with OR and compared ECOG/WHO score 2 or over to less than 2, whereas ECOG/WHO score 1 compared 0 did not increase the risk for short-term mortality [66] (S1 Table and S6 Appendix).

Two studies could not be included in the meta-analysis and one reported that performance status was significantly associated with short-term mortality [51], whereas the other one reported that performance status was not significantly associated with short-term mortality [59].

### Frailty, functional status, quality of life or fatigue

One included study found that the association between lower MMSE (<26) or ADL (<6) and overall survival was non-statistically significant, but the study did not provide prognostic estimates for MMSE and ADL in their model [108]. Authors from one study provided additional information on the prognostic value of age, performance status and comorbidity on QoL measured by the QLQ-C30 among 97 participants aged 60 years or more that they had not reported in their article [98]. The analysis found that patients 60–69 years old have

**Table 2. Grading of recommendations, assessment, development, and evaluation—confidence in estimates of effect for quality of life.**

| Number of studies | Quality assessment | | | | | | Effect | Quality |
|---|---|---|---|---|---|---|---|---|
| | Study design | Risk of bias | Inconsistency | indirectness | Imprecision | Other considerations | Description | |
| Age (60–69 years old versus 70+ years) | | | | | | | | |
| 1 | Observational studies | Serious | Not serious | Not serious | Very serious | None | No statistical association (p = 0.135) | ⊕○○○ Very low |
| Karnofsky performance score (2-point change) | | | | | | | | |
| 1 | Observational studies | Serious | Not serious | Not serious | Serious | None | QLQ-C30 (95% CI: 1 to 4) | ⊕⊕○○ Low |
| Comorbidities measured by diagnosis count (per 1 unit) | | | | | | | | |
| 1 | Observational studies | Serious | Not serious | Not serious | Very serious | None | No statistical association | ⊕○○○ Very low |

QLQ-C30: European Organization for the Research and Treatment of Cancer (EORTC) 30-item questionnaire.

worse QoL than those 70+ years old, but the interaction between age and follow up time was not statistically significant (p = 0.135), which means that there is no statistically significant difference in change in QoL over time between the age groups. The study also found that a 10-point change in the Karnofsky performance score is associated with a 2-point change on QLQ-C30 (95% CI: 1 to 4). There was no statistical association between comorbidities measured by diagnosis count (per 1 unit) and QoL (Table 2). We did not find eligible studies that reported the prognostic values of these factors on frailty and functional status.

## Discussion

In this review, we included 90 studies and found age, comorbidities, and performance status are associated with an increase in long-term and short-term mortality. Among these factors, we found high and moderate quality evidence for age (per increase in 5-year increments, 70 + years versus <70 years old, 75+ years versus <75 years), comorbidities (HCT-CI: score 3 or over VS less than 3) and performance status (ECOG/WHO: score 2 or over VS less than 2) on long-term mortality. Therefore, older patients with advanced age (especially 70 years or old), more comorbidities (HCT-CI score 3 or over) and poor performance status (ECOG/WHO score 2 or over) are at a higher risk of long-term mortality. The limited evidence available on QOL and lack of evidence on frailty and functional status restrict our understanding of the prognostic values of factors on these outcomes.

The results from our review informed the development of guidelines for the management of older adults with newly diagnosed AML. There were no important changes in conclusions when comparing the evidence that the guideline panel considered when formulating their recommendations and the full body of evidence through October 2021 described in this manuscript. Clinicians using the recommendations can use our results to identify which older patients are considered appropriate for antileukemic therapy. Clinicians can use the tools and thresholds for which we found high and moderate quality evidence (e.g., age: 70 or 75 years, HCT-CI: score 3, ECOG/WHO: score 2) to do pre-antileukemic therapy assessment and engage in shared decision-making with patients (e.g., choose therapy or encourage participation in clinical trials) to improve outcomes in this population. Furthermore, clinicians or researchers should consider that QoL or fatigue are important outcomes for patients, and frailty or function impairment is common among older AML patients [110]. Future studies

with low risk of bias focused on these factors or outcomes are needed to support such assessments.

Despite the fact that cytogenetics remains relevant in disease biology in patients with AML, older patients usually have unfavourable cytogenetics, and intermediate or favourable risk cytogenetics does not truly exist in older patients [67, 111]. Therefore, we did not include this predictor in our systematic review. However, some cytogenetic abnormalities may provide prognostic values as some new emerging personalized treatments may target some gene mutations or biomarkers [112]. The quality of evidence varied across outcomes, predictors, and methods of measurement. Although most of the studies addressed the relationship between the predictors and mortality, the amount of evidence available for the influence of these predictors on QoL was limited.

We found that the impact of age on increasing mortality was marginally greater in those receiving more intensive therapy (HR 1.21) than in those receiving less intensive therapy (HR 1.08) and was also marginally greater in studies in which the first patient was enrolled before 2000 (HR 1.28) than those recruited after 2000 (HR 1.13). These findings suggest that the increased risk of dying with older age may depend on what chemotherapy regimen one is receiving. It thus further suggests that as new therapies are developed the association of age with mortality may change. In particular, new non-intensive but effect treatments may blunt some of the association between age and mortality.

Our review is the first systematic review addressing the prognosis value of age, comorbidities, and performance status on patient-important outcomes (mortality, QoL or fatigue) among older newly diagnosed AML patients who receive antileukemic therapy. The scope of our review was determined by a panel of experts who formally prioritized the most important questions related to prognosis in their practice. We followed high methodological standards for conducting our review. We used the GRADE approach to assess the certainty in the evidence and interpret the results, which allows for making conclusions that consider all important aspects in the body of evidence.

Because studies reported the prognosis value using various tools or thresholds, the number of studies included in the meta-analysis for some predictors are relatively small. We could not pool data about the prognosis value of interested factors on short-term mortality for included studies used different cut-off points or categories of predictors. Due to reporting limitations, unfortunately, many studies could not be included in the meta-analysis. For example, 17 studies only reported P values or statistical significance, or did not provide sufficient information about how they categorized age [23, 24, 36, 37, 39, 51, 57, 60, 61, 64, 73, 76, 80, 93–95, 97]. Seven of them reported that age was not statistically associated with long-term mortality [36, 44, 57, 73, 80, 84, 94]. Although this may be perceived as being inconsistent with the results of our study, the lack of statistical significance may be due to lack of power of such studies. We are not supervised that their estimates were inconsistent when compared to those of the studies we were able to include in meta-analyses. Similarly, there were 2 included studies without enough information to be included in the meta-analysis that reported no statistical association between performance status and long-term mortality [59, 80]. We assume these studies would not change our results and conclusions substantially. Furthermore, we are lack information on the mechanisms for why these prognostic factors impact long-term mortality. The possible mechanisms may include higher risks of sepsis, need for ICU admission, early relapse, or not achieving remission and so on.

In conclusion, high-quality or moderate-quality evidence support that age, comorbidities, performance status predicts the prognosis of older patients with AML undergoing antileukemic therapy.

## Supporting information

**S1 Appendix. Search strategy.**
(DOCX)

**S2 Appendix. Characteristics of included studies.**
(DOCX)

**S3 Appendix. Risk of bias assessment.**
(DOCX)

**S4 Appendix. Forest figures for age.**
(DOCX)

**S5 Appendix. Forest figures for comorbidity.**
(DOCX)

**S6 Appendix. Forest figures for performance score.**
(DOCX)

**S7 Appendix. Subgroup analysis and funnel plots.**
(DOCX)

**S1 Table. SOF for short-term mortality.**
(DOCX)

**S1 Checklist.**
(DOC)

**S1 Data.**
(XLSX)

## Acknowledgments

This systematic review was performed as part of the ASH clinical practice guidelines on AML among older people. The authors thank all the guideline panel members who do not in the author list for their advice and suggestions. The authors also thank Dr. Shane Gangatharan at Department of Haematology, Royal Perth Hospital, Australia, Dr. Sacha Satram-Hoang at Q. D. Research, Inc, USA and Drs. Alan K. Burnett and Robert Hills at Department of Haematology, School of Medicine, Cardiff University, UK. for providing additional data or information. The authors also thank Dr. Toshiaki A. Furukawa from Kyoto University for dealing with publications in Japanese.

## Author Contributions

**Conceptualization:** Romina Brignardello-Petersen.

**Data curation:** Qiukui Hao, Mi Ah Han, Tahira Devji, Fernando Kenji Nampo, Shabbir M. H. Alibhai.

**Formal analysis:** Qiukui Hao, Farid Foroutan.

**Investigation:** Qiukui Hao, Fernando Kenji Nampo, Sudipto Mukherjee, Shabbir M. H. Alibhai, Ashley Rosko, Mikkael A. Sekeres.

**Methodology:** Qiukui Hao, Farid Foroutan, Mi Ah Han, Tahira Devji, Gordon H. Guyatt, Romina Brignardello-Petersen.

**Resources:** Sudipto Mukherjee, Shabbir M. H. Alibhai, Ashley Rosko, Mikkael A. Sekeres, Gordon H. Guyatt, Romina Brignardello-Petersen.

**Supervision:** Romina Brignardello-Petersen.

**Writing – original draft:** Qiukui Hao.

**Writing – review & editing:** Farid Foroutan, Mi Ah Han, Tahira Devji, Fernando Kenji Nampo, Sudipto Mukherjee, Shabbir M. H. Alibhai, Ashley Rosko, Mikkael A. Sekeres, Gordon H. Guyatt, Romina Brignardello-Petersen.

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
