## [Decision Letter · Decision Letter 0]

4 Aug 2022

PONE-D-21-40244Prognosis of older patients with newly diagnosed AML undergoing antileukemic therapy: a systematic reviewPLOS ONE

Dear Dr. Hao,

Thank you for submitting your manuscript to PLOS ONE. After careful consideration, we feel that it has merit but does not fully meet PLOS ONE’s publication criteria as it currently stands. Therefore, we invite you to submit a revised version of the manuscript that addresses the points raised during the review process.

We look forward to receiving your revised manuscript.

Kind regards,

Ahmet Emre Eşkazan, M.D.

Academic Editor

PLOS ONE

Journal Requirements:

2. Thank you for stating the following financial disclosure: "NA" 

3. Thank you for stating the following in your Competing Interests section: "None"

Reviewers' comments:

Reviewer's Responses to Questions

**Comments to the Author**

1. Is the manuscript technically sound, and do the data support the conclusions?

Reviewer #1: Yes

Reviewer #2: Yes

2. Has the statistical analysis been performed appropriately and rigorously? 

Reviewer #1: Yes

Reviewer #2: I Don't Know

3. Have the authors made all data underlying the findings in their manuscript fully available?

Reviewer #1: Yes

Reviewer #2: Yes

4. Is the manuscript presented in an intelligible fashion and written in standard English?

Reviewer #1: Yes

Reviewer #2: Yes

5. Review Comments to the Author

Reviewer #1: This is a good effort to perform a meta analysis to define contributing factors to long term mortality in older adults with AML. While the results are no surprise and not substantially different than the data available to treating physicians, the efrort to pull in published data and assess quality of evidence is a good effort.

1. Authors need to clarify how they reconciled the subgroup effect of treatment and treatment time period on age effect on mortality. It may be critical as with new non intensive but effective treatments may blunt some of the effects of age on mortality

2. The authors have made minimal reference to not using cytogenetics as their contention is favorable or intermediate risk AML does not exist in older population. THis can be generally true but again with recent advances in effective therapies show that cytogenetics or disease based risk categories do matter. Authors need to highlight this potential shortcoming in discussion

3. Statement in line 389 needs to be modified "Clinicians using the recommendations can use our results to identify which older patients are considered appropriate for antileukemic therapy". I would rather use the results presented to choose therapy or encourage participation in clinical trials so that we can improve outcomes in this high risk population and not to discourage therapy.

4. Authors claim that their analysis links QoL with age, comorbidities but the number of studies are limited. This needs to be acknowledged

Reviewer #2: Your research has been reviewed according to PRISMA 2020 Guideline.

The title is appropriate.

Introduction

The rationale for the review is described well, but it is recommended to write an introductory paragraph on quality of life and fatigue in AML and it can be changed with the paragraph between lines 77-83.

Objectives are clearly defined.

“We conducted a … of older patients with AML” should be stated in the objectives.

Methods

Inclusion and exclusion criteria for the review are specified clearly, but you should also state how studies were grouped for the syntheses in this paragraph.

Information sources, search strategy, selection process and data collection, data items, methods used to access the risk of bias, effect measures, synthesis methods, bias assessment, and certainty assessment are specified.

Results

Study selection is described, and studies are cited.

Characteristics of studies are presented.

The risk of bias in studies is presented.

Results of the individual studies and results of syntheses are given appropriately.

For each synthesis, the risk of bias is presented.

Certainty of evidence is presented.

Discussion

The discussion provides a general interpretation of the results and discusses the limitations of the paper.

6. PLOS authors have the option to publish the peer review history of their article (what does this mean?). If published, this will include your full peer review and any attached files.

Reviewer #1: No

Reviewer #2: **Yes: **Tayfun Elibol

---

## [Author Response · Author response to Decision Letter 0]

30 Sep 2022

Please see the attached document, thanks.

---

## [Decision Letter · Decision Letter 1]

21 Nov 2022

Prognosis of older patients with newly diagnosed AML undergoing antileukemic therapy: a systematic review

PONE-D-21-40244R1

Dear Dr. Hao,

We’re pleased to inform you that your manuscript has been judged scientifically suitable for publication and will be formally accepted for publication once it meets all outstanding technical requirements.

Kind regards,

Ahmet Emre Eşkazan, M.D.

Academic Editor

PLOS ONE

Additional Editor Comments (optional):

Reviewers' comments:

Reviewer's Responses to Questions

**Comments to the Author**

1. If the authors have adequately addressed your comments raised in a previous round of review and you feel that this manuscript is now acceptable for publication, you may indicate that here to bypass the “Comments to the Author” section, enter your conflict of interest statement in the “Confidential to Editor” section, and submit your "Accept" recommendation.

Reviewer #1: All comments have been addressed

Reviewer #2: All comments have been addressed

2. Is the manuscript technically sound, and do the data support the conclusions?

Reviewer #1: Yes

Reviewer #2: Yes

3. Has the statistical analysis been performed appropriately and rigorously? 

Reviewer #1: Yes

Reviewer #2: Yes

4. Have the authors made all data underlying the findings in their manuscript fully available?

Reviewer #1: Yes

Reviewer #2: Yes

5. Is the manuscript presented in an intelligible fashion and written in standard English?

Reviewer #1: Yes

Reviewer #2: Yes

6. Review Comments to the Author

Reviewer #1: Authors have responded appropriately to the reviewers' comments. The current version can be accepted

Reviewer #2: All the suggestions we have given have been properly implemented.

Congratulations and wish you continued success.

7. PLOS authors have the option to publish the peer review history of their article (what does this mean?). If published, this will include your full peer review and any attached files.

Reviewer #1: No

Reviewer #2: **Yes: **Tayfun Elibol

---

## [Editor Report · Acceptance letter]

22 Nov 2022

PONE-D-21-40244R1 

Prognosis of older patients with newly diagnosed AML undergoing antileukemic therapy: a systematic review 

Dear Dr. Hao:

I'm pleased to inform you that your manuscript has been deemed suitable for publication in PLOS ONE. Congratulations! Your manuscript is now with our production department. 

Kind regards, 

on behalf of

Dr. Ahmet Emre Eşkazan 

Academic Editor

PLOS ONE